# The Expression and Function of *Notch* Involved in Ovarian Development and Fecundity in *Basilepta melanopus*

**DOI:** 10.3390/insects15040292

**Published:** 2024-04-19

**Authors:** Yifei Xie, Yifan Tan, Xuanye Wen, Wan Deng, Jinxiu Yu, Mi Li, Fanhui Meng, Xiudan Wang, Daohong Zhu

**Affiliations:** 1Laboratory of Insect Behavior and Evolutionary Ecology, College of Life and Environmental Sciences, Central South University of Forestry and Technology, Changsha 410004, China; xyf8779@163.com (Y.X.); tanyifan1105@163.com (Y.T.); m18866961204@163.com (F.M.); 2Institute of Forestry and Grassland Protection, Hunan Academy of Forestry, Changsha 410018, China; dengwan@hnlky.cn (W.D.); yujinxiu2000@hnlky.cn (J.Y.); limi49@163.com (M.L.); 3Center for Biological Disaster Prevention and Control, National Forestry and Grassland Administration, Shenyang 110031, China; wenxuanye_1116@163.com

**Keywords:** *Basilepta melanopus*, leaf beetle, *notch* gene, RNA interference, pest management

## Abstract

**Simple Summary:**

We identified the *notch* gene in the reproductive system of the *Basilepta melanopus* beetle, a pest affecting oil tea plants. It was discovered that the Notch signaling pathway is crucial for the early development of insect ovaries. Specifically, suppressing the Notch gene expression by RNAi led to impaired ovarian development, decreased mating rates, and reduced egg production. These findings highlight the potential of RNAi-based, Notch-targeted pest control strategies as a practical approach for managing the forest pest *B. melanopus* populations.

**Abstract:**

*Basilepta melanopus* is a pest that severely affects oil tea plants, and the Notch signaling pathway plays a significant role in the early development of insect ovaries. In this study, we explored the function of the *notch* gene within the Notch signaling pathway in the reproductive system of *B. melanopus*. The functional domains and expression patterns of *Bmnotch* were analyzed. *Bmnotch* contains 45 epidermal growth factor-like (EGF-like) domains, one negative regulatory region, one NODP domain and one repeat-containing domain superfamily. The qPCR reveals heightened expression in early developmental stages and specific tissues like the head and ovaries. The RNA interference (RNAi)-based suppression of *notch* decreased its expression by 52.1%, exhibiting heightened sensitivity to ds*Notch* at lower concentrations. Phenotypic and mating experiments have demonstrated that ds*Notch* significantly impairs ovarian development, leading to reduced mating frequencies and egg production. This decline underscores the Notch pathway’s crucial role in fecundity. The findings advocate for RNAi-based, Notch-targeted pest control as an effective and sustainable strategy for managing *B. melanopus* populations, signifying a significant advancement in forest pest control endeavors.

## 1. Introduction

The Notch signaling pathway is one of the key pathways regulating the early formation of insect oocytes. The critical protein in this pathway, Notch, is evolutionarily conserved and has a transmembrane domain. It is located on the cell membrane of anterior follicle cells and binds to the Delta ligand, which is also a transmembrane protein found on adjacent germ cells [1]. Subsequently, the signal is amplified by the highly expressed glycosyltransferase Fringe and ultimately cleaved by the γ-secretase complex (presilin and nicastrin) to release the Notch intracellular domain (NICD). NICD enters the nucleus and binds to the hairless inhibitor (Su(H)), mediating changes in the expression of downstream target genes (e.g., E(spl)) in specific tissues and developmental stages [2]. The Notch signaling pathway relies primarily on interactions between the Notch protein and various ligands to transmit signals between cells and plays a vital role in reproduction and development processes in insects [3]. In studies of the fruit fly *Drosophila melanogaster*, Notch was shown to mediate the functional differentiation of cells involved in various physiological activities, such as the formation of organ precursor cells [4], humoral and cellular immunity [5,6], cell proliferation [7], and the formation of germ cells in the testes and ovaries [8,9]. In several insects, Notch pathway genes influence the normal formation of follicle cells, thereby controlling the production of normal egg chambers [10,11,12].

*Basilepta melanopus* is an important forest pest. This insect primarily feeds on the tender leaves and new shoots of the oil tea plant *Camellia oleifera*. The interconnected boreholes on tea leaves, resulting from feeding damage, lead to withering, leaf dropping, and overall tree health decline. *B. melanopus* exhibits a high feeding capacity, strong reproductive ability, and considerable migratory capability. During peak infestation periods, it poses a substantial threat to oil tea production [13]. Currently, it is widely distributed in southeastern provinces such as Fujian, Guangdong, Guangxi, Jiangxi, and Hunan. Controlling *B. melanopus* relies heavily on chemical pesticides, which, in turn, raise numerous environmental and ecological concerns. Targeting the reproductive behavior of the pest has been proven to be an effective pest control method. However, there is currently limited research on the functional aspects of reproductive-related genes in this insect species.

With the advancement of science and technology, RNA interference (RNAi) technology has shown great potential in the field of pest control. This technique employs precise intervention in genes that are crucial to the vital activities of pests, thereby controlling aspects of their growth, immunity, resistance, metabolism, reproduction, and chemical sensing [14]. Through the sequencing of insect genomes, researchers have identified a large number of pest-specific genes and specific gene sequences, which are key to the success of RNAi-based pest control strategies [15]. Currently, significant advancements have been achieved in the realm of genetic pest control, such as Lepidoptera, Coleoptera, Orthoptera, Diptera, Hemiptera, and Isoptera, by interfering with the expression of their key genes within laboratory settings [16,17,18,19,20,21]. In summary, RNA interference technology provides a new and efficient method for pest control, utilizing molecular biology techniques to directly target the life processes of pests, demonstrating significant potential in pest management and control.

In this study, we cloned the *notch* gene of the oil tea leaf beetle, predicted its functional regions and analyzed the temporal and spatial expression patterns. Then, we explored its role in ovarian development, mating success, and oviposition behavior in *B. melanopus*, laying the foundation for the development of targeted biological control technologies based on RNAi.

## 2. Materials and Methods

### 2.1. Insects

Insects for the experiment were originally collected from Guiyang County, Chenzhou, Hunan Province, China, and were cultured in conical flasks placed in a light incubator. The culture conditions were maintained at a temperature of 27 ± 1 °C, a photoperiod of 14 h of light and 10 h of dark (14 L:10 D), and a relative humidity of 80%.

### 2.2. Sample Collection and RNA Extraction

The eggs, larvae, pupae, and adults were collected. The collected *B. melanopus* adults were dissected for various tissues, including the head, foregut, midgut, hindgut, ovaries, and testes, with 30 mg of tissue each composing one biological replicate. The ovaries of female insects were dissected and collected 1, 3, 5, and 7 days after eclosion. The tissue samples were temporarily stored in RNA preservation liquid during the dissection process. Samples from different developmental stages and tissues were immediately frozen in liquid nitrogen and then used for total RNA isolation using the Steady Pure Universal RNA Extraction Kit (Accurate Biology, Changsha, China), following the instructions provided in the manual. Three biological replicates were prepared: 4 individuals as one biological replicate in the stages and 20 individuals as one biological replicate in the tissue sample.

### 2.3. Bioinformatics Analyses

The *notch* gene was identified in the *B. melanopus* transcriptome (PRJNA1071291) by tBLASTn using homologous sequences from *T. castaneum*. For preliminary domain function predictions, the genes were analyzed via InterProScan 99.0 in EMBL-EBI (https://www.ebi.ac.uk/interpro/ (accessed on 23 January 2024)). A comprehensive set of 35 *notch* gene sequences from various insect species was retrieved from the GenBank database (Appendix A). These sequences were meticulously aligned using CLUSTAL X 1.83 to facilitate the construction of a neighbor-joining phylogenetic tree, with 1000 bootstrap replicates to ensure robustness and reliability utilizing MEGA 4.1. Utilize Jalview version 2 (https://www.jalview.org/ (accessed on 26 February 2024)) for the multiple sequence alignment of the *notch* amino acid sequences among the *B. melanopus*, *D. melanogaster*, *T. castaneum*, *Bombyx mori*, and *Aedes aegypti*. The full-length cDNA sequence has been submitted to the NCBI GenBank, with the accession number PP254332.

### 2.4. Temporal and Spatial Expression Analysis

The sizes of the RNA bands were assessed via 1% agarose gel electrophoresis, and the quality and concentration of the RNA were measured using a K5600 spectrophotometer (Kaiao Technology Co., Beijing, China). The RNA samples were reverse-transcribed using the PrimeScript^TM^ RT reagent Kit (Takara, Beijing, China) following the instructions for cDNA synthesis. Real-time quantitative PCR was carried out using a CFX96 Real-Time System (Bio-Rad, Hercules, CA, USA) with the SYBR Green *ProTaq* qPCR Kit (Takara, China). *Notch* gene-specific primers and primers for fluorescent quantitative PCR were designed using Primer 3 (http://primer3.ut.ee/ (accessed on 6 April 2023)) (Appendix A). qPCR detection was performed in a 20 μL reaction containing 10 μL 2× SYBR green *ProTaq* HS Premix, 1 μL cDNA template (100 ng), 0.4 μL of each primer, 0.4 μL ROX Reference Dye and 7.8 μL nuclease-free H_2_O. The cycling program was 95 °C for 2 min, 40 cycles at 95 °C for 15 s, 60 °C for 30 s, 60 °C for 15 s and 95 °C for 30 s. The relative expression level was calculated using the 2^−ΔΔCt^ method [22], with *β-actin* as the reference gene [13]. Fold changes were determined after the relative expression values were standardized using the lowest value. One-way analysis of variance was used to statistically analyze the data, and the means were separated using a least significant difference test at a significance level of *p* < 0.05 using SPSS 16 (SPSS, Chicago, IL, USA).

### 2.5. dsRNA Synthesis

The dsRNAs were synthesized using the T7 RiboMAX^TM^ Express RNAi System (Promega, Madison, WI, USA). The DNA templates were obtained by PCR amplification using primers containing T7 RNA polymerase promoter sequences (Appendix A). The dsRNAs for enhanced green fluorescent protein (EGFP), ds*EGFP*, served as negative controls. Subsequently, ds*Notch* and ds*EGFP* were prepared according to the instructions in the manual, and the purified dsRNAs were dissolved in nuclease-free water. The purity and integrity of the dsRNAs were analyzed using a spectrophotometer and 1% agarose gel electrophoresis before usage.

### 2.6. RNA Interference

Adult insects (one-day post-eclosion) were subjected to a 3 h starvation period. The prepared 500/1000/2000 ng/μL ds*Notch* solutions were painted onto the leaf surfaces (4 cm^2^ per leaf), achieving ds*Notch* concentrations of 5/10/20 ng/cm² per leaf, respectively. The ds*EGFP* solution of 500 ng/μL was applied in the control. Each adult was fed 4 leaves per 24 h. This feeding regimen was continued for 48 h, after which the insects were fed untreated leaves for an additional 48 h. The treatment groups were exposed to three concentrations of ds*Notch*, 500 ng/μL, 1000 ng/μL, or 2000 ng/μL, with 20 insects selected for each concentration. RNA extraction was performed on 5 insects (3–4 g) from each group on the 2nd, 4th, and 8th days after initiation of dsRNA feeding. RNAi efficiency was assessed by qPCR.

In a separate set of experiments, 10 female adults were starved for three hours and then placed in an enclosure. These insects were then fed *C. oleifera* leaves soaked in 1000 ng/μL ds*Notch* for 48 h, followed by a diet of untreated leaves. On the 4th day after the initiation of dsRNA feeding, ovarian dissection was performed, and RT-qPCR was used to analyze the changes in the expression of the *Bmnotch*. On the 6th day, the ovaries were dissected for phenotypic observation. The experiments above were conducted in triplicate, with ds*EGFP* serving as the control group.

To enhance mating efficiency, 40 male and 30 female insects were selected for the mating experiment. The females (two days post-eclosion) were fed either 1000 ng/μL ds*Notch* or ds*EGFP*. After a period of 5 days, these females were adapted to untreated males in a breeding enclosure. Mating occurrences were observed and recorded throughout the day. The differences in mating rates between the treatment group and the control group were statistically analyzed using a chi-square test. Following the completion of mating in both groups, the insects were further reared to record and compare the final egg-laying quantities.

## 3. Results

### 3.1. Bioinformatics Analyses of the Notch Gene in B. melanopus

The functional domains of the *notch* amino acid sequence were analyzed using InterProScan 99.0, which revealed 45 epidermal growth factor-like (EGF-like) domains (spanning amino acids 30 to 1425), one negative regulatory region (NRR, 1453–1577 aa), one NODP domain (1654–1719 aa), one ankyrin repeat-containing domain superfamily (1870–2122 aa, containing 7 ankyrin repeat domains) and a PEST domain (C-terminal, 2389–2440 aa). Within the EGF-like domains, eight calcium-binding conserved sites were identified. The NRR domain is composed of three cysteine-rich Lin12-NOTCH repeats (LNRs) (Figure 1A).

Utilizing amino acid sequences sourced from NCBI, we analyzed the phylogenetic relationships of *Bmnotch* with those of other insects. The *notch* genes of Coleopteran insects formed a distinct clade, with *B. melanopus* exhibiting the closest phylogenetic affinity to *Leptinotarsa decemlineata*, a species of leaf beetle commonly associated with Solanaceous crops (Figure 1B). Multiple sequence alignments indicate that *Bmnotch* exhibits a high similarity with the NRR within the selected model insect *notch* genes (Figure 1C).

### 3.2. Temporal and Spatial Expression of Bmnotch

The temporal and spatial expression of *Bmnotch* was measured via qPCR. During various developmental stages, the expression level of *Bmnotch* was significantly higher in eggs, larvae, and pupae than in adults, with an expression multiple of 4.4 to 4.9 (Figure 2A). Among the different tissues, the expression level of *Bmnotch* was the highest in the head, followed by that in the ovaries, with significantly greater expression in these two segments and tissues than in the other tissues. Specifically, the expression level in the head was 11.48 times greater than that in the Malpighian tubules, where it was the lowest, and the expression level in the ovaries was 5.94 times greater than that in the Malpighian tubules (Figure 2B). In the ovaries on different days, *Bmnotch* expression was lowest on day 7. The expression on day 3 was significantly greater than that on days 5 and 7, with expression levels on days 1, 3, and 5 being 2.4, 4.3, and 1.3 times greater than that on day 7, respectively (Figure 2C).

### 3.3. Silencing of Bmnotch and Its Effect on Ovary Development

RNAi was conducted using three different concentrations of ds*Notch*, and the results demonstrate a significant reduction in expression levels on the second day at 2000 ng/µL, on the fourth day at 1000 ng/µL, and on the eighth day at 1000 ng/µL, with silencing efficiencies of 42.5%, 50.0%, and 49.1%, respectively (Figure 3A). RNAi was conducted using ds*Notch* at a concentration of 1000 ng/µL, and ovaries extracted 8 days post-treatment were analyzed. The results indicate a significant decrease in expression levels, with a silencing efficiency of 52.1% (Figure 3B).

Observation of ovaries dissected from 6 day post-RNAi treatment reveal that ovaries from the ds*Notch* treatment group were smaller and less developed compared to the control group (Figure 4A). Mating rates were determined on the 5th and 6th days, revealing that on the 5th day, the mating rates for the treatment and control groups were 20% and 33.3%, respectively. On the 6th day, these rates changed to 3 μL 7.5% for the treatment group and 75% for the control group, demonstrating a significant difference (Figure 4B). Egg production was quantified 48 h post-mating, with the treatment and control groups producing 63 and 121 eggs, respectively. The number of eggs produced increased to 91 for the treatment group and 162 for the control group at 72 h post-mating (Figure 4C).

## 4. Discussion

In *D. melanogaster*, 36 EGF-like repeats, seven ankyrin repeat domains, one PEST domain and one NRR domain were identified in the *notch* gene. EGF-like repeats primarily engage in ligand binding, and their number and arrangement impact the interaction between Notch and its ligands, thereby affecting signaling pathway activation and specificity. Ankyrin repeats, located intracellularly, mediate interactions between Notch and other internal proteins, facilitating signal transmission from the membrane to the nucleus and activating gene transcription. The PEST domain contains signals for rapid protein degradation, thus controlling the intensity and duration of the Notch signaling response [23,24]. In the absence of ligand binding, the LNR (Lin12-Notch repeats) domain conceals the metalloprotease site (S2 site), thereby preventing unwarranted S2 cleavage. This function of the LNR domain is crucial for regulating the activation of the Notch signaling pathway. This ensures that the Notch receptor is activated only when appropriate, maintaining the specificity and efficiency of the pathway [25]. In this study, we identified the *notch* gene in the *B. melanopus* transcriptome database and characterized its protein sequence. Phylogenetic analysis further confirmed the identity of the retrieved sequence (Figure 1). Our analysis indicated that *Bmnotch* contains all key functional domains of the *notch* gene; these domains are involved primarily in signal transduction functions and potentially play a significant role in biological processes such as cell proliferation in *B. melanopus*.

RNAi in insects is a complex system, with multiple factors influencing the outcomes of RNAi treatments, including the mechanisms of dsRNA uptake and the systemic nature of RNAi within these organisms. A key factor is the degradation of dsRNA by dsRNase in the gut and hemolymph, leading to reduced interference efficiency [17,26,27]. However, some Coleopteran insects are more sensitive to dsRNA, with the ingestion of small amounts leading to significant phenotypic and gene expression reduction [28,29,30]. For instance, feeding larvae of *Plagiodera versicolora* with leaves treated with 8 ng/cm² dsRNA results in a 100% mortality rate [31]. In our study, *Bmnotch* showed a 50% reduction in expression after treatment with 1000 ng/μL ds*Notch*, similar to the silencing effects observed in other beetles. At a concentration of 2000 ng/μL, only a 50% silencing efficiency was observed on day 2, after which the effect was not significant, indicating a lack of sensitivity of the beetle to dsNotch at this concentration. This phenomenon may be related to the complexity of the RNAi system, which we will investigate further in further research.

The *notch* gene and its signaling pathway play crucial roles in various developmental processes in insects, including wing formation, embryonic morphogenesis, and regionalization [32,33,34,35]. Studies on the *notch* gene in the brown planthopper indicate that its expression is greater in adults than in pupae, with elevated expression in the wings and midgut compared to other tissues [33]. In the present study, the expression levels of *Bmnotch* were found to be greater in eggs, larvae, and pupae than in adults, suggesting the potential importance of the gene during these three stages. Additionally, tissue expression analysis during the adult stage indicated that the head and ovaries may be the primary functional tissues for the *Bmnotch* protein. The expression level of *Bmnotch* in ovaries is higher than that in other age groups, and its temporal expression profile correlates with early ovarian maturation, suggesting that it plays a potential role in ovarian development. The differential spatial and temporal expression patterns of the *notch* gene in the planthopper and *B. melanopus* indicate functional variations across different insect species. In this study, we acknowledge the possibility that spatial expression patterns differ at other developmental stages, necessitating further research for more precise conclusions.

During the ovarian development of insects, the principal follicle cells undergo three cellular cycles: mitosis, the endocycle, and gene amplification [36]. The Notch pathway plays a pivotal role in regulating the transition of follicle cells from mitosis to the endocycle [10,37]. In the case of *A. aegypti*, the injection of dsRNA targeting *notch* and its downstream gene JNK significantly reduces the production of normal egg granules in female mosquitoes, thereby controlling the population of the next generation [11]. In *Blattella germanica*, the absence of the *notch* gene leads to abnormal changes in the late stages of follicle cell development [12]. The interference of *notch* in *Tribolium castaneum* causes follicle cells to prematurely enter the endocycle while immature, disrupting the formation and patterning of the egg chamber and leading to a reduction in the number of terminal filament cells and stalk cells [10]. After interference with *notch* in *B. melanopus*, the expression of *Bmnotch* in the ovaries decreases, and a subsequent delay in ovarian development occurs (Figure 4A). These conditions likely contribute to the reduced mating rate observed in the reproductive capability tests of the treated group, which in turn could be a potential cause of reduced oviposition.

*Notch* regulates the synthesis of certain proteins involved in the growth and segmentation of antennae. For instance, disrupting the expression of *notch* leads to a decrease in the density of sensory bristles on the antennae [38]. Similarly, RNAi of the downstream gene nub in the Notch pathway also results in the absence of sensory bristles [39]. The RNAi-mediated knockdown of *notch* in the antennae of silkworms causes the significant fusion of antennal segments and milder defects, such as the fusion of lateral branches [40]. The antennae of insects play a crucial role in mating behaviors, particularly in sex recognition and attraction, especially in detecting pheromones and other chemical signals [41,42]. Therefore, the regulation of antennal growth by the *notch* gene may affect insect mating behaviors. These findings provide a new perspective on the regulatory role of *notch* in reproductive behavior. In our research, *Bmnotch* was highly expressed in the head, but whether *Bmnotch* regulates the growth process of antennae and thereby affects mating rates requires further experimental verification.

In summary, disrupting the expression of *notch* blocks or disturbs signal transduction via the Notch pathway, causing severe obstacles in the formation of insect reproductive cells, thus preventing normal reproductive activities and leading to population decline. Therefore, in-depth research into the key genes involved in the Notch signaling pathway in insects could provide a theoretical basis for the development of novel insecticides by exploiting their potential as targets.

## Figures and Tables

**Figure 1 insects-15-00292-f001:**
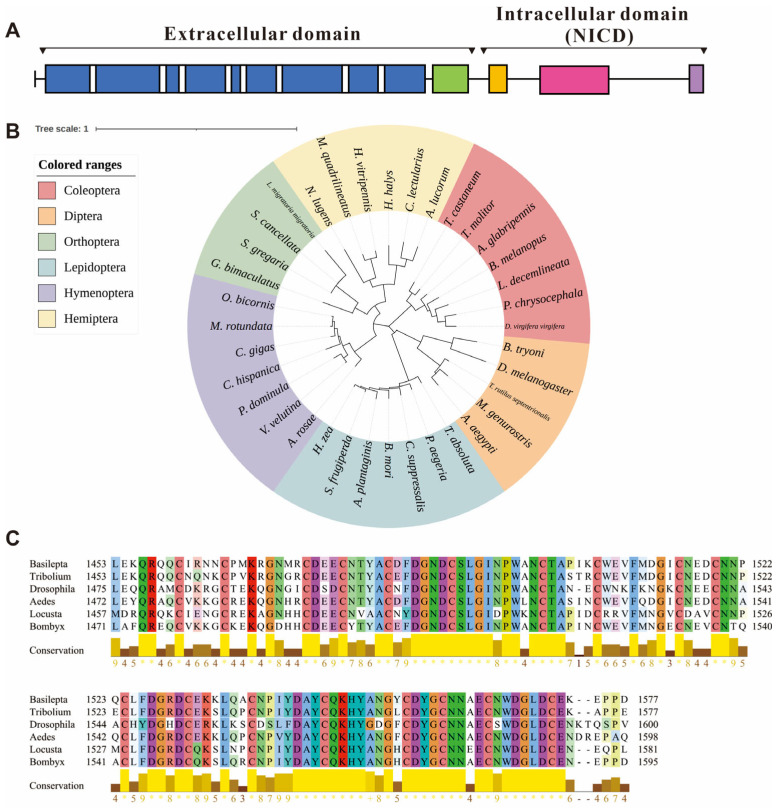
Information of the *notch* gene in *Basilepta melanopus*. (**A**) Preliminary domain prediction based on the Notch amino acid sequence. Blue, epidermal growth factor-like (EGF-like) domains; green, negative regulatory region (NRR); orange, NODP domain; pink, ankyrin repeat-containing domain superfamily; purple, PEST domain. (**B**) Phylogenetic analysis of *notch* in a range of insect species. The neighbor-joining algorithm was used to construct the phylogenetic tree. (**C**) Multiple sequence alignments of the *notch* amino acid sequences in negative regulatory region (NRR) among the *B. melanopus*, *D. melanogaster*, *T. castaneum*, *Bombyx mori*, and *Aedes aegypti*.

**Figure 2 insects-15-00292-f002:**
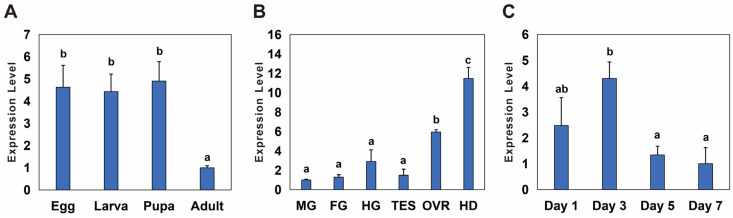
Spatiotemporal expression profiles of *notch* in *Basilepta melanopus*. (**A**) The differential temporal expression patterns of *Bmnotch* across various developmental stages, specifically, eggs, larvae, pupae, and adults. (**B**) The relative spatial expression patterns of *Bmnotch* across a spectrum of tissues and anatomical segments, including the midgut (MG), foregut (FG), hindgut (HG), ovary (OVR), testis (TES), and head (HD). (**C**) The dynamic changes in the expression of *Bmnotch* across ovaries of varying ages. Samples were procured from the ovaries of female adults at the ages of 1, 3, 5, and 7 days. The expression levels of *Bmnotch* were quantified by qPCR. The relative expression levels were computed relative to the minimum observed value, which was standardized to a baseline of 1. Statistical disparities among the groups were evaluated using one-way analysis of variance (ANOVA) followed by a post hoc LSD test, with the mean values presented alongside the standard error (SE) (*p* < 0.05). Different lowercase letters indicate significant differences between the two groups of data.

**Figure 3 insects-15-00292-f003:**
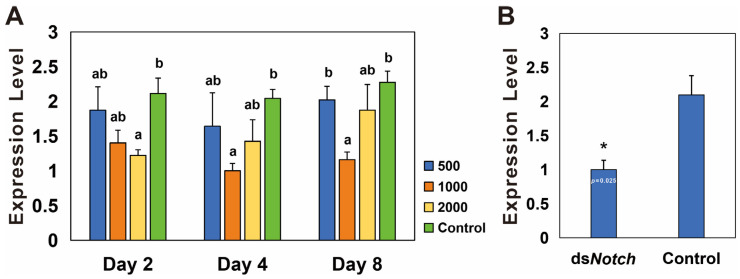
Expression levels of *Bmnotch* after RNAi knockdown. (**A**) Differential silencing efficacy of ds*Notch* at various concentrations on *Basilepta melanopus*. Actin served as the reference gene, and ds*EGFP* was utilized as the control. The experiment was replicated three times, and the data were analyzed via the one-way analysis of variance (*p* < 0.05). Error bars represent the SEM. (**B**) Silencing efficiency of *notch* in *B. melanopus* ovaries 4 days after initiation of RNAi. A *t*-test was used to test for significant differences (*p* < 0.05). Different lowercase letters or asterisk indicate significant differences between the two groups of data.

**Figure 4 insects-15-00292-f004:**
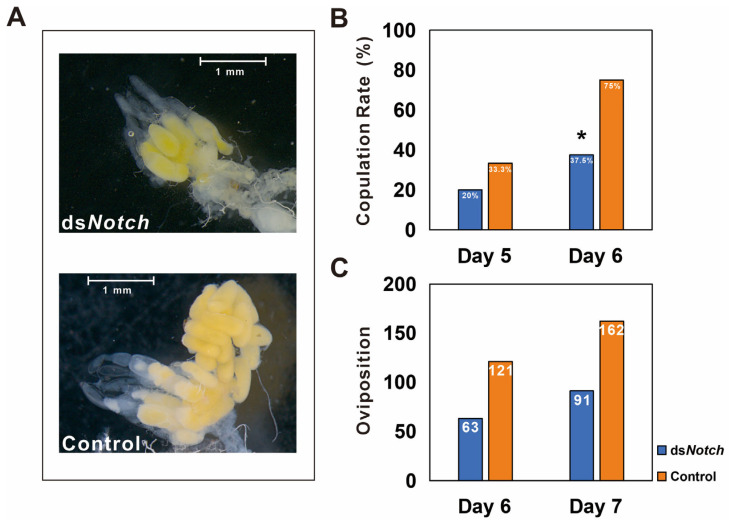
Phenotype and reproductive capacity test of *Basilepta melanopus* females after *Bmnotch* RNAi. (**A**) Phenotype of fractional ovaries after *Bmnotch* RNAi. (**B**) Copulation rate of females treated with ds*Notch* paired with normal males. ds*EGFP* was used as a control. (**C**) The total number of eggs was calculated for the mated couples. Asterisk indicate significant differences between the two groups of data.

## Data Availability

Data are contained within the article and Appendix A.

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
