# Peer review of "The Expression and Function of Notch Involved in Ovarian Development and Fecundity in Basilepta melanopus"

_insects, 2024, doi:10.3390/insects15040292_

Round 1

Reviewer 1 Report

Comments and Suggestions for Authors

Major issues:

1.     In Materials and Methods, while the authors mentioned the number of insects used for some experiments, it would be better to provide the sample size (n) for all experiments, including n for both treatments and data analyses. For example, in Figure 3 and Figure 4, the n number for data analysis needs to be given. For 2.6 RNA interference, the size of leaves needs to be described (line 142), and the rationale for choosing 40 males and 30 females needs to be explained (the ratio why not to be set to 1:1) (line 155). Besides, the volume of dsRNA used needs to be given as well.

2.     For gene silencing efficiency, the authors carried out the RNAi experiments with different dsRNA concentrations (500, 1000, and 2000 ng/µL, respectively). However, the silencing efficiency seemed to be similar, with a range of 42%–50%. After the knockdown of Notch (at dsRNA concentrations of 500 ng/µL and 2000 ng/µL), the expression of Notch was not decreased when compared to control (Figure 3). Can the authors explain the reason? Is this caused by gene off-target or RNAi approach? Please discuss in the Discussion.

3.     In Figure 4, the authors presented the phenotype of the ovary without any description in the main text. So, what information Figure 4A can bring to the readers? Please provide descriptive information in the results.

4.     In the Discussion, the authors discussed the study of the brown planthopper, a hemipteran insect, however, phylogenetic analysis of Notch excluded Hemiptera. Therefore, the brown planthopper is not a good example for discussion here. Since the literature on the Notch signaling pathway is available in many insect species, I would suggest discussing more studies on the insect species that were analyzed in the phylogenetic tree and removing all relevant text concerning the brown planthopper.

Minor issues:

Line 15: Italicize “Basilepta melanopus”. Similar issue in line 273.

Line 129: Change “P<” to “P <”. And make sure the written form is united in the manuscript (italic or not).

Lines 122, 123, 145, and l50: Change “µl” to “µL”. Make sure the written form of the unit is in a united format in the manuscript. Please check thoroughly the manuscript to update the remaining issues that are not mentioned here.

Line 151: Change “RT–qPCR” to “RT-qPCR”

Lines 167–170: Change short hyphen “-” to en dash “–”.

Line 336: Change “drosophila” to “Drosophila”. Similar issues in lines 342, 346, 349, and 353. Please update all the genus names for other insects with similar issues in the reference list. Similar issues for RNAi, dsRNA, RNA, and PCR. Please carefully check all title names of the listed literature in the reference list.

For gene name, when writing after “ds” which refers to gene and should be written in italics. Please update all written forms in the manuscript.

In Table S1, point out the T7 sequence.

Reviewer 2 Report

Comments and Suggestions for Authors

The authors attempted to understand the roles of notch gene in the reproductive system of the Basilepta melanopus beetle, a pest affecting oil tea plants. This study presents a comprehensive analysis of the notch's function, expression patterns, and the potential of RNA interference (RNAi)-based Notch-targeted pest control strategies. In general, the experiments are well designed, the results are clear, and the discussion is nice. I just have a few minor comments as the follows.
1)Line 15: Notch gene, Throughout the document, the term "notch" is used inconsistently regarding capitalization and italic. It is recommended to standardize the use of "Notch" when referring to the gene or its protein product for consistency and to align with scientific naming conventions.

2)Line 15-16: Basilepta melanopus, Latin scientific names should be italicized.

3) Line 111: Jalview, Please provide detailed information about the software.

4) Line 158: 40 male and 30 female insects, Please inform us about the instar stage of these adults.

5) Line 212-213: quantitative polymerase chain reaction, This description can be abbreviated as qPCR.

6) Line 216: least significant difference, This description can be abbreviated as LSD.

7) Line 242: Figure 4B, It would be better to present the copulation rate in each bar.

8) Line 279: B. melanopus, Latin scientific names should be italicized.

Reviewer 3 Report

Comments and Suggestions for Authors

In this study, the authors investigated the function of the notch gene within the Notch signaling pathway in the reproductive system of Basilepta melanopus. Moreover, the authors analyzed the functional domains and expression patterns of the Notch gene. The results showed that essential domains of notch, involved in cell proliferation and signal transduction, exhibit heightened expression in early developmental stages and specific tissues like the head and ovaries. The RNA interference (RNAi) based suppression of Notch decreased its expression by 52.1%, markedly impairing ovarian development and reducing mating rates and egg production. This decline underscores the Notch pathway's crucial role in fecundity.

Overall, it is a very interesting study. The data is sufficient to support the conclusion of this study, and the results are of interest. However, the manuscript needs careful proofreading and revision. Grammar mistakes are undermining the significance of this study. Besides, the authors need to address the following comments, which will make this document more strong.

-              The authors need to remove the grid lines from all figures.

-              The abstract is too short and simple. The authors need to add some detailed, important results. Sometimes, the readers want to read the abstract to get important results, so authors need to add some details about their important findings.

-              Some figures are colorful, while some are black and white. The authors need to make a consistency. All figures should be colorful or black and white.

-              Figure 4A (Phenotype of fractional ovaries after Bmnotch RNAi) is not very clear. If possible, please provide a more attractive and detailed figure

-              The discussion is too narrow, the authors need to interpret their results in more detail and justify it with the recently published literature

Comments on the Quality of English Language

The manuscript needs careful proofreading and revision. Grammar mistakes are undermining the significance of this study

Round 2

Reviewer 3 Report

Comments and Suggestions for Authors

The authors have addressed all comments with full justifications. Therefore, I recommend this manuscript for publication in insects